# Transcription shapes genome-wide histone acetylation patterns

Benjamin J. E. Martin [1], Julie Brind'Amour [2], Anastasia Kuzmin[1], Kristoffer N. Jensen[2], Zhen Cheng Liu[1], Matthew Lorincz [2] & LeAnn J. Howe [1✉]

Histone acetylation is a ubiquitous hallmark of transcription, but whether the link between histone acetylation and transcription is causal or consequential has not been addressed. Using immunoblot and chromatin immunoprecipitation-sequencing in *S. cerevisiae*, here we show that the majority of histone acetylation is dependent on transcription. This dependency is partially explained by the requirement of RNA polymerase II (RNAPII) for the interaction of H4 histone acetyltransferases (HATs) with gene bodies. Our data also confirms the targeting of HATs by transcription activators, but interestingly, promoter-bound HATs are unable to acetylate histones in the absence of transcription. Indeed, HAT occupancy alone poorly predicts histone acetylation genome-wide, suggesting that HAT activity is regulated post-recruitment. Consistent with this, we show that histone acetylation increases at nucleosomes predicted to stall RNAPII, supporting the hypothesis that this modification is dependent on nucleosome disruption during transcription. Collectively, these data show that histone acetylation is a consequence of RNAPII promoting both the recruitment and activity of histone acetyltransferases.

[1] Department of Biochemistry and Molecular Biology, Life Sciences Institute, Molecular Epigenetics Group, University of British Columbia, 2350 Health Sciences Mall, Vancouver, BC, V6T 1Z3 Canada. [2] Department of Medical Genetics, Life Sciences Institute, Molecular Epigenetics Group, University of British Columbia, 2350 Health Sciences Mall, Vancouver, BC, V6T 1Z3 Canada. ✉email: ljhowe@mail.ubc.ca

Lysine acetylation of histone amino-terminal tails has been linked to gene expression for many decades[1]. More recently, genome-wide localization studies across eukaryotes, including yeast and mammals, revealed that histone tail acetylation primarily occurs at the promoters and 5′ ends of transcribed genes[2]. Although some forms of acetylation have been referred to as "global" and "non-targeted"[3], genome-wide occupancy studies show that histone acetylation levels correlate strongly with transcription, suggesting a causal relationship between the two.

Histone acetylation is catalyzed by conserved histone acetyltransferases (HATs), generally consisting of a catalytic subunit complexed with auxiliary proteins required for enzymatic activity and targeting[4]. Most HAT complexes have relatively low substrate specificity and modify multiple lysine residues within either H3 or H4. Thus, histone acetylation sites within H3 and H4 generally show similar distributions[2], and mutations of histone lysine residues, with the exception of H4K16, result in comparable changes in gene expression[5]. Histone acetylation is a dynamic mark due to the activity of histone deacetylase complexes (HDACs). Similar to HATs, HDACs generally exist as multi-protein complexes with catalytic subunits that can deacetylate multiple lysines on one or more histones[6].

In Saccharomyces cerevisiae, the most well-characterized proteins with lysine acetyltransferase activity are Gcn5 and Esa1, which are the catalytic subunits of multiple HAT complexes, including the H3-specific HATs, SAGA, and ADA for Gcn5, and the H4-specific HATs, NuA4, and Piccolo for Esa1[4]. SAGA and NuA4 are targeted to gene promoters via an interaction between a shared subunit, Tra1, and DNA-bound transcription activators[7–9], which is thought to target acetylation of nucleosomes flanking promoters. This, together with the observation that Gcn5 and Esa1 are required for transcription of multiple genes[10–13], has led to the widely accepted model that histone acetylation acts upstream of transcription initiation. It should be noted however that, in addition to histones, HATs acetylate many non-histone proteins involved in transcription initiation[14]. As such, whether SAGA and NuA4 activate transcription primarily through acetylation of core histones remains uncertain.

While the current model for targeting HATs by transcription activators upstream of transcription is widely accepted, there are several examples of histone acetylation being deposited as a consequence of transcription. Numerous HATs interact with co-transcriptional H3K4me3[15–18] and the phosphorylated carboxy-terminal domain (CTD) of RNAPII[12,19]. In addition, co-transcriptional histone exchange mediates incorporation of acetylated histones into nucleosomes within transcribed regions[20], and recent work has shown that RNA can promote the activity of CBP at enhancers[21]. Taken together these observations suggest that histone acetylation can also be a consequence of the transcription process. However, the relative contribution of these pathways to histone acetylation patterns remains unknown.

In this study, we sought to determine the relative contributions of the "causal" vs. "consequential" pathways for targeting histone acetylation to transcribed genes. We first found that inhibition of transcription results in rapid histone deacetylation in both yeast and mouse embryonic stem cells (mESCs), demonstrating that a significant portion of histone acetylation is a consequence of transcription. Loss of RNAPII also results in depletion of Epl1, a subunit of the yeast NuA4 and Piccolo HATs, from gene bodies, consistent with HAT targeting by RNAPII. Residual, transcription-independent Epl1 binding was observed over promoter regions, in agreement with the targeting of HATs by transcription activators, but surprisingly, acetylation in these regions is still transcription-dependent. Thus, although these results are consistent with previous models for targeting HATs, we found that HAT recruitment alone is insufficient to mediate

acetylation of the associated nucleosomes, indicative of post-recruitment regulation of acetyltransferase activity. One model that is consistent with our data is that the acetylation of histones is dependent on nucleosome disruption by RNAPII. In agreement with this, we see increased acetylation at nucleosomes predicted to impede RNAPII passage.

## Results

**Histone acetylation is dependent on transcription.** Despite the well-known correlation between histone acetylation and transcription, whether this posttranslational modification (PTM) is primarily a cause or consequence of transcription has not been definitively tested. We, therefore, sought to assess the dependence of histone acetylation on transcription by inhibiting RNAPII activity in S. cerevisiae. Previous studies have used the rpb1-1 temperature-sensitive mutant to disrupt transcription[22]. However, more recent experiments have suggested that this mutant does not directly inhibit RNAPII, as shifting the mutant to the restrictive temperature has minimal effects on transcript synthesis[23] and does not lead to rapid dissociation of RNAPII from gene bodies[24]. To achieve effective inhibition of RNAPII, we, therefore, treated cells with 1,10 phenanthroline monohydrate (1,10-pt), which has been shown to rapidly inhibit transcript synthesis[23]. Confirming efficient transcription inhibition by 1,10-pt, we observed a global loss of RNAPII serine 5 CTD phosphorylation by immunoblot analysis (Fig. 1a) and rapid alterations in RNAPII distribution as determined by ChIP-seq (Fig. 1b, ChIP inputs shown in Supplementary Fig. 1a). Immunoblot analysis of yeast whole-cell extracts showed that within 15 m of transcription inhibition, a broad range of H3 and H4 acetylation marks were rapidly lost (Fig. 1a, c). Similar deacetylation was observed following, treatment with the transcription inhibitor thiolutin (Supplementary Fig. 2a), or degradation of Rpb2, the second largest subunit of RNAPII, using an auxin-inducible degron (Supplementary Fig. 2b). Notably, loss of acetylation was dependent on the histone deacetylases Rpd3 and Hda1 (Supplementary Fig. 2c) and could be almost completely blocked by prior treatment with the HDAC inhibitor TSA (Supplementary Fig. 2d), confirming active deacetylation upon transcription inhibition. Histone acetylation loss was due to disruption of HAT activity, rather than increased HDAC activity, as incubation with TSA following 1,10-pt treatment failed to restore histone acetylation (Supplementary Fig. 2e). HATs are conserved throughout Eukaryota and thus it is likely that acetylation is dependent on transcription in other organisms. Indeed, we found that inhibition of transcription by actinomycin D, which inhibits transcription initiation, as indicated by loss of Ser5p (Fig. 1d), and transcription elongation[25] in mESCs resulted in the loss of H3K9ac and H3K27ac in bulk histones (Fig. 1d, e).

To confirm that histone acetylation loss is a direct consequence of transcription inhibition, we treated S. cerevisiae cells with 1,10-pt for 15 m and performed ChIP-seq for H3K23ac, H4K8ac, and H4K12ac. Consistent with previous studies[26,27], we used non-transcribed regions to account for global changes in ChIP-seq experiments (see "Methods"). While no major changes to nucleosome occupancy or position were observed following the short transcription inhibition performed here, large decreases in histone acetylation were observed (Fig. 1f). Importantly, heat-maps of $\log_2$ fold changes in ChIP-seq signal upon 1,10-pt treatment showed that patterns of histone acetylation loss mirrored those of RNAPII (Supplementary Fig. 3a). In addition, histone deacetylation was limited to nucleosomes that lost RNAPII upon 1,10-pt treatment (Supplementary Fig. 3b). In contrast, regions with more stable RNAPII, including the 3′ ends of genes, showed slight increases in histone acetylation (Fig. 1f,

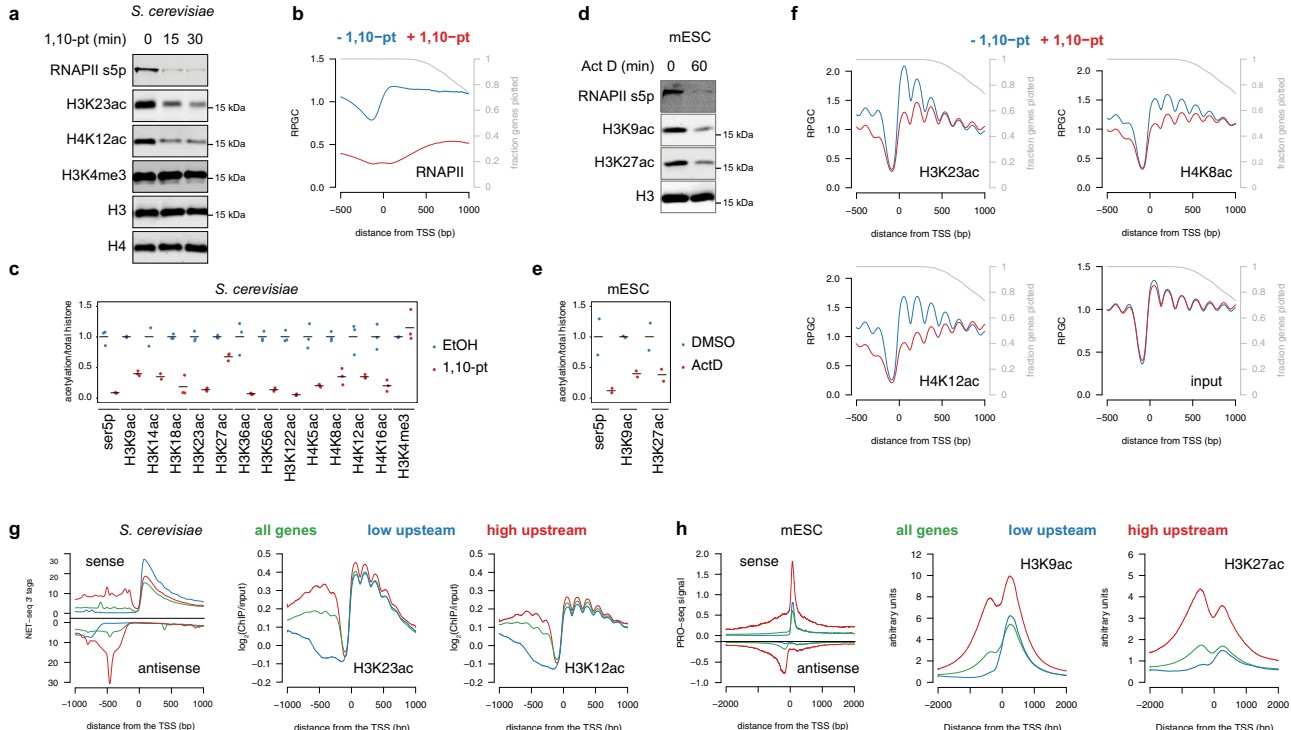

**Fig. 1 The majority of histone acetylation is dependent on transcription. a** Whole-cell extracts from *S. cerevisiae* cells before and after treatment with 1,10-pt were subjected to immunoblot analysis with the indicated antibodies. Experiments were performed in triplicate with quantified results shown in (**c**). Raw immunoblot data is provided in the Source Data file. **b** Average profile of RNAPII (Rpb3 ChIP-seq[73]) at 5206 transcribed genes aligned by the TSS before (blue) and after a 15 m treatment with 1,10-pt (red). Only data until the polyadenylation site (PAS) was included, and the gray line represents the fraction of genes still being plotted. RPGC reads per genomic coverage, TSS transcription start site. **c** Strip plots of histone PTM immunoblot signals normalized to histone H4 levels from three independent yeast whole-cell extracts from cultures without (blue) and with (red) a 15-min treatment with 1,10-pt. Horizontal lines indicate the mean, with the vehicle control set to 1. **d** Nuclear extracts from mouse ESCs before and after treatment with actinomycin D were subjected to immunoblot analysis with the indicated antibodies. Experiments were performed in duplicate with quantified results shown in (**e**). Raw immunoblot data is provided in the Source Data file. **e** Strip plots of histone acetylation immunoblot signals normalized to histone H3 from two mESC nuclear extracts from independent cultures without (blue) and with (red) actinomycin D. Horizontal lines indicate the mean, with the vehicle control set to 1. **f** Average profile of H3K23ac, H4K8ac, and H4K12ac (MNase ChIP-seq) and input at 5206 genes aligned by the TSS before (blue) and after (red) a 15-min treatment with 1,10-pt. Data from 1,10-pt-treated cells was normalized to untreated (see "Methods"). **g** The average signal relative to the TSS for *S. cerevisiae* NET-seq[41], H3K23ac[33], and H4K12ac ChIP-seq data relative to all genes (green) or 832 transcribed genes that have low (blue) or high (red) upstream NET-seq signal. The ChIP data is presented as log$_2$(ChIP/input) as nucleosome density is not consistent between the different gene bins. **h** The average signal relative to the TSS for PRO-seq[74], and H3K9ac and H3K27ac ChIP-seq data relative to all genes (green) or 3035 transcribed genes that have low (blue) or high (red) upstream PRO-seq signal.

Supplementary Fig. 3a, b). While this may result from enhanced targeting of HATs displaced from other loci, we cannot rule out the possibility that our scaling approach did not fully account for global decreases in histone acetylation. Irrespective, we find that H3K23ac, H4K8ac, and H4K12ac were primarily deacetylated at regions that lost RNAPII upon transcription inhibition, suggestive of a direct effect. Collectively these results demonstrate that a large portion of histone acetylation is a consequence of transcription, which is inconsistent with the prevalent model that histone acetylation is primarily targeted to active genes upstream of transcription. Although this result was initially surprising, it is consistent with reports demonstrating that histone acetylation upstream of promoters is limited to those with divergent transcription[28–30], which was confirmed by analyses of data from both yeast (Fig. 1g, Supplementary Fig. 4) and mESCs (Fig. 1h).

**Piccolo is targeted by RNAPII.** The simplest explanation for the transcription dependence of histone acetylation is that RNAPII targets HATs to transcribed genes. Indeed, previous work has shown that genome-wide occupancies of Gcn5, Sas3, and Epl1, a common subunit of Esa1-dependent HATs[31], are increased at highly transcribed genes[27,32,33]. To directly test this, we performed ChIP-seq for Epl1 prior to and following 15 m of transcription inhibition. In the absence of 1,10-pt, Epl1 was enriched over gene bodies and depleted immediately upstream of TSSs (Fig. 2a, inputs shown in Supplementary Fig. 1b). Following transcription inhibition, Epl1 occupancy over the 5′ genic regions diminished significantly, mirroring RNAPII loss in these regions (compare Figs. 2a and 1b, Supplementary Fig. 5a, b). In contrast, Epl1 bound upstream of TSSs was less sensitive to transcription inhibition (Fig. 2a), and analysis of genes lacking divergent transcription to avoid signal of Epl1 on upstream gene bodies, showed a peak of Epl1 binding ~400 bp upstream of TSSs in both actively transcribing and transcription-inhibited cells (Fig. 2b, inputs shown in Supplementary Fig. 1c). Collectively, these results suggest that Epl1 is recruited to chromatin through two pathways: transcription-dependent targeting to gene bodies and transcription-independent binding upstream of TSSs.

Epl1 is a component of two HATs: NuA4 and Piccolo[31]. Both complexes contain a HAT module consisting of Esa1, Epl1, and Yng2, but only NuA4 contains Tra1, which mediates the interaction of this HAT with transcription activators[7,9]. Deletion

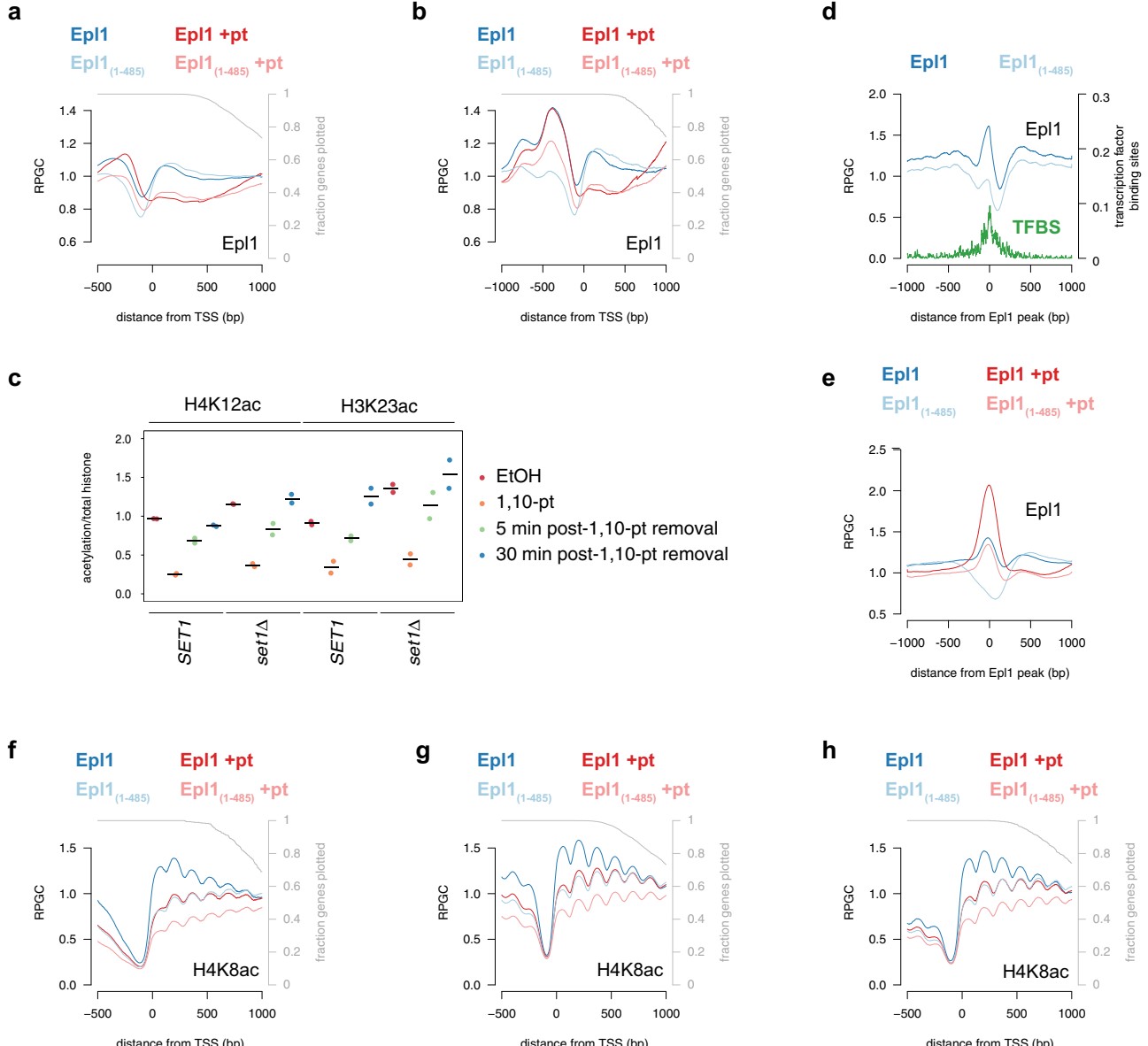

**Fig. 2 Transcription promotes the interaction of H4-specific HATs with chromatin. a, b** Average profile of Epl1 (dark lines) or Epl1$_{1-485}$ (light lines) ChIP-seq from sonicated extracts at all genes (5206) (**a**) or 832 unidirectional promoter genes (**b**) aligned by the TSS before (blue) and after (red) a 15-min treatment with 1,10-pt. Only data until the PAS was included, and the gray line represents the fraction of genes plotted for each position. Data from drug-treated and mutant cells were normalized to untreated wild-type (see "Methods"). RPGC reads per genomic coverage, TSS transcription start site. **c** Strip plots of histone acetylation immunoblot signals normalized to histone H4 levels from two independent yeast whole-cell extracts. Wild type and set1Δ cells were either untreated (red) or treated with 1,10-pt for 30 m, followed by TSA treatment for an additional 30 m (orange), before being washed into fresh media containing TSA. Samples were collected 5 (green) and 30 (blue) minutes post wash. Horizontal lines indicate the mean. **d** Average profile of Epl1 (dark blue) and Epl1$_{1-485}$ (light blue) ChIP-seq from MNase-treated chromatin relative to the center of 562 regions showing strong Epl1 peaks (peak coordinates are provided in the Source Data file). The abundance of transcription factor binding sites (TFBS)[38] across the region is shown in green. **e** Average profile of Epl1 (dark lines) and Epl1$_{1-485}$ (light lines) ChIP-seq from sonicated extracts from cells before (blue) and after (red) a 15-min treatment with 1,10-pt, relative to the center of 562 regions showing strong Epl1 peaks. **f–h** Average profiles of H4K8ac ChIP-seq from MNase-digested extracts from Epl1 (dark lines) and Epl1$_{1-485}$ (light lines)-expressing strains, before (blue) and after (red) a 15-min treatment with 1,10-pt, at genes with strong Epl1 peaks (**f**), all genes (**g**), or 832 genes with promoters lacking divergent transcription (**h**).

of the 348 C-terminal amino acids of Epl1 (Epl1$_{1-485}$), disrupts the incorporation of the HAT module into NuA4[31], and thus cells expressing Epl1$_{1-485}$ are thought to have Piccolo, but not NuA4. We used this information to determine whether bound Epl1 was in the form of NuA4 or Piccolo by repeating ChIP-seq in cells expressing Epl1$_{1-485}$. Analysis of all and just unidirectional promoters (Fig. 2a, b) showed that truncation of Epl1 resulted in a loss of binding upstream of TSSs, with little change over gene

bodies, which is consistent with Piccolo and NuA4 binding gene bodies and promoters respectively.

RNAPII-dependent targeting of Esa1 to gene bodies was proposed to occur through recognition of H3K4 methylation by the PHD finger of Yng2[16,34,35]. However, Fig. 1a shows that H3K4me3 was resistant to transcription inhibition, indicating that acetylation loss was not due to the removal of this PTM. Moreover, genome-wide analysis of HAT occupancy suggests

that, while H3K4 methylation can promote HAT processivity, it is dispensable for targeting[27]. To determine whether H3K4 methylation is important for transcription-dependent acetylation, wild-type and *set1Δ* strains were treated with 1,10-pt, which was then washed out to allow cells to resume transcription prior to the assessment of acetylation levels by immunoblot. Figure 2c shows that acetylation was restored with similar kinetics in wild-type and mutant strains. Thus, the ability of Piccolo to acetylate bulk nucleosomes was not dependent on the presence of H3K4 methylation following transcription resumption. It should be noted, however, that because we analyzed bulk histones, we cannot rule out the possibility that loss of H3 methylation altered the ability of Esa1 to acetylate nucleosomes at specific regions, but this loss is masked by increased acetylation in other regions.

Previous work has also implicated RNAPII serine 5 phosphorylation and H3K36 methylation in targeting NuA4 to gene bodies[12,35,36]. While this is inconsistent with our genome-wide data showing that disruption of NuA4 has minimal impact on Epl1 binding to these regions, using *set2Δ* (Supplementary Fig. 6a) and analog-sensitive *KIN28* (Supplementary Fig. 6b, c) mutants, we show that global histone acetylation was not dependent on H3K36 methylation nor RNAPII serine 5 phosphorylation by Kin28, either alone or in combination with loss of H3K4 methylation. Thus, although the exact mechanism for targeting Epl1 to gene bodies remains to be identified, these results show that the transcription-dependence of histone H4 acetylation can be partially explained by a requirement of transcription for targeting Piccolo to gene bodies.

**NuA4 is targeted by transcription activators.** To confirm that Epl1 bound upstream of TSSs represents activator targeted HAT complex, we sought to improve the resolution of our Epl1 ChIP-seq data. To this end, we repeated Epl1 ChIP-seq in actively transcribing cells using chromatin fragmented by micrococcal nuclease, which was previously shown to detect nonhistone protein complexes bound to DNA[37]. Epl1-bound, MNase-resistant DNA showed a similar pattern of genome-wide localization as input chromatin (Supplementary Fig. 1d, e), although Epl1 and Epl1$_{(1-485)}$, but not untagged ChIP-sequence fragments, showed increased abundance on highly expressed genes (Supplementary Fig. S1f–h). Using a high stringency cut-off (see methods), we identified 562 promoters with strong peaks of Epl1 (Fig. 2d, inputs shown in Supplementary Fig. 1i). These peaks originated from sub-nucleosome sized DNA fragments that did not precipitate with anti-acetyl-histone antibodies (Supplementary Fig. 7) and thus were unlikely to represent nucleosomes. NDRs with Epl1 peaks were wider (mean of 297 vs. 164 bp, *p* value of two-sided students *t* test $2.0 \times 10^{-49}$) and associated with more highly expressed genes (*p* value of two-sided students *t* test

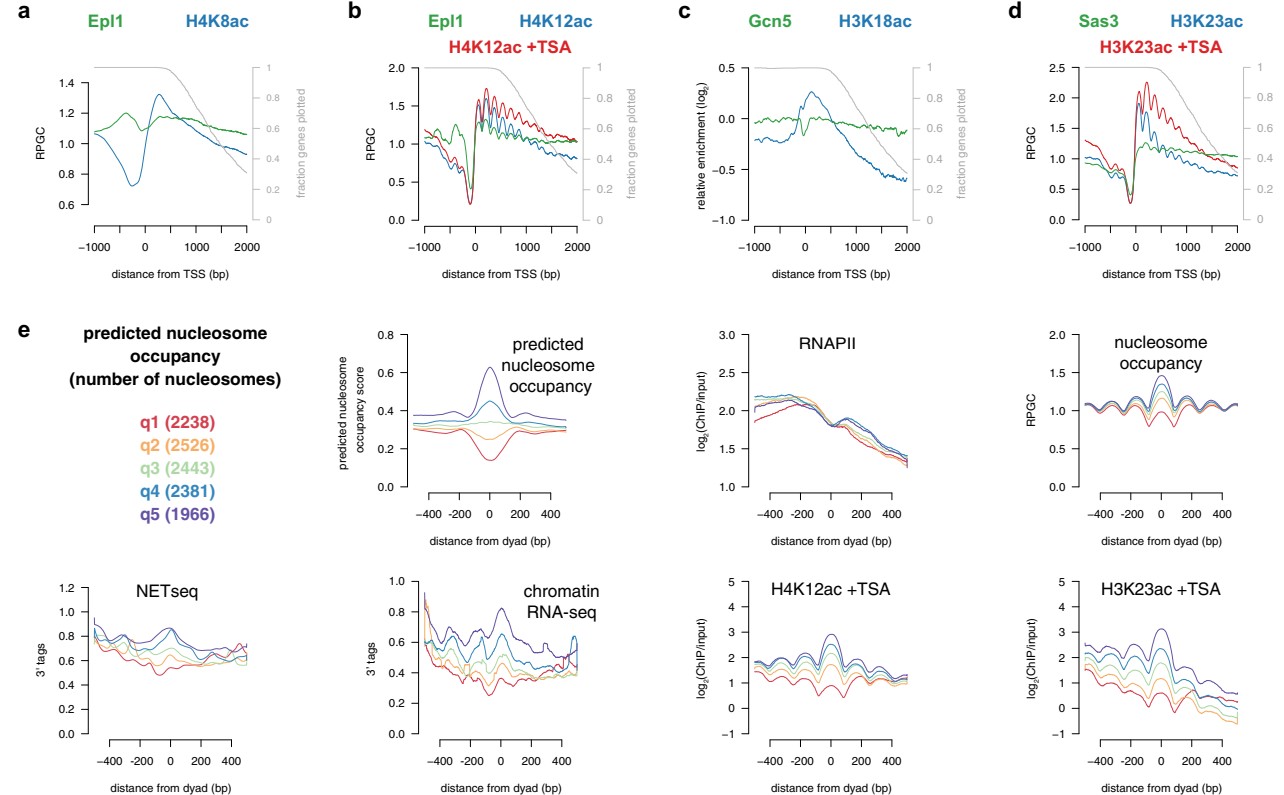

**Fig. 3 The activity of histone acetyltransferases are regulated post-recruitment. a**, **b** Average profiles of Epl1 (green) and H4 acetylation (blue) ChIP from sonicated[27] (**a**) or MNase-digested chromatin (**b**) at 832 unidirectional promoter genes aligned by the TSS. Only data until the PAS was included, and the gray line represents the fraction of genes plotted for each position. Data from TSA-treated cells (red) was normalized to untreated wild-type (see "Methods"). RPGC reads per genomic coverage, TSS transcription start site. **c**, **d** Average profiles of Gcn5 (green) and H3K18ac (blue) ChIP[75] from sonicated chromatin (**c**) and Sas3 (green) and H3K23ac (blue) ChIP from MNase-digested chromatin (**d**) at 832 unidirectional promoter genes aligned by the TSS. **e** Genic nucleosomes[47] in the middle quintile for RNAPII occupancy [log$_2$(Rpb3/input) 50 bp upstream and downstream of the nucleosome dyad] were divided into quintiles based on predicted nucleosome occupancy[48] over the same region. Shown are profiles upstream and downstream of the nucleosome dyad for predicted nucleosome occupancy, RNAPII (Rpb3 ChIP-seq[73]), nucleosome occupancy (MNase-seq), NET-seq[41], chromatin RNA-seq[41], and nucleosome-normalized H4K12ac and H3K23ac[33] ChIP-seq from TSA-treated cells for the first (red), second (orange), third (green), fourth (blue), fifth (purple) quintiles for predicted nucleosome occupancy.

$3.5 \times 10^{-70}$) than all NDRs. Importantly, these peaks overlapped regions with transcription factor binding sites[38], and Epl1 occupancy in these regions was reduced following Epl1 truncation (Fig. 2d), consistent with NuA4 targeting by transcription activators. A comparison of NDRs with Epl1 peaks to NDRs depleted for Epl1 signal revealed strong enrichment of DNA binding motifs for several transcription factors (Supplementary Fig. 8a). For the two top hits, Rap1 and Aft2, the DNA binding motifs were enriched at and immediately upstream of the Epl1 peak centers (Supplementary Fig. 8b, c), supporting a model of direct recruitment of Epl1, and consistent with reported Rap1 targeting of NuA4[9].

Analysis of sonicated ChIP-seq data over Epl1-enriched promoters confirmed that Epl1 binding to these regions was independent of transcription but dependent on the C-terminus of Epl1 (Fig. 2e, inputs in Supplementary Fig. 1j). Moreover, the truncation of Epl1 resulted in a loss of H4K8 acetylation on nucleosomes immediately downstream of TSSs on the Epl1-enriched genes (Fig. 2f, inputs in Supplementary Fig. 1k) consistent with NuA4-dependent acetylation in these regions. Surprisingly, however, the presence of HATs at promoters was insufficient to trigger histone acetylation on its own. Analysis of cells treated with 1,10-pt showed reduced H4K8ac in both wild-type and Epl1 mutants (Fig. 2f), indicating that regardless of the mode of HAT targeting, acetylation remained dependent on transcription. Similar results were observed when analyzing all genes (Fig. 2g, inputs in Supplementary Fig. 1l) or genes lacking divergent transcription (Fig. 2h, inputs in Supplementary Fig. 1m). Collectively, these results confirm the targeting of HATs by transcription activators, but also show that Epl1-dependent acetylation of 5′ nucleosomes requires RNAPII activity.

**Histone acetyltransferases are regulated post-recruitment**. The requirement of transcription for localization of Piccolo to gene bodies implies that Epl1 occupancy should mirror histone H4 acetylation marks. However, analysis of genes with unidirectional promoters, to avoid the confounding effect of divergent transcription, revealed Epl1 bound throughout gene bodies, while H4 acetylation was primarily enriched in 5′ regions [Fig. 3a (sonicated ChIP-seq) and b (MNase ChIP-seq), MNase inputs shown in Supplementary Fig. 1n]. Similar results were observed following treatment of cells with TSA (Fig. 3b), and thus the differences between HAT occupancy and acetylated histone levels were not due to HDACs reshaping acetylation patterns. A similar disconnect was observed between the occupancies of the H3-specific HATs, Gcn5, and Sas3, and histone H3 acetylation (Fig. 3c, d, input for 3d shown in Supplementary Fig. 1o). Thus, we observe a poor correspondence between HAT localization and histone acetylation, which is consistent with previous suggestions that regulation of HAT activity, following chromatin binding, is a major determinant of histone acetylation genome-wide[21,27,33,39].

High-resolution mapping of engaged RNAPII by NET-seq, CRAC-seq, and chromatin-bound RNA-seq, shows RNAPII accumulation at the 5′ ends of genes[28,40,41], which is proposed to represent either slow passage or premature transcription termination of RNAPII in these regions. Consistent with the latter, 5′ accumulation is not observed with PRO-seq[42], which measures elongation competent RNAPII. The accumulation of histone acetylation at the 5′ ends of genes, together with the transcription-dependence of histone acetylation, suggests that the presence of RNAPII promotes the activity of HATs. In vitro and in silico studies demonstrate that histone tails are tightly associated with DNA, making them poor substrates for HATs[43–46], but nucleosome disruption during RNAPII passage

could displace tails from DNA, promoting histone acetylation by available HATs. If this hypothesis is correct, then the longer RNAPII spends traversing a nucleosome, the greater is the chance that the histones will be acetylated. To test this, we asked whether nucleosomes that are more likely to impede RNAPII exhibit increased histone acetylation in vivo. To control for total RNAPII levels, annotated gene body nucleosomes[47] were divided into five bins based on Rpb3 ChIP-seq signal [$\log_2$(ChIP/input)]. Each bin was further split into quintiles based on the predicted ability to strongly or weakly form nucleosomes[48]. This approach enabled the identification of nucleosomes with similar levels of RNAPII but differing predicted nucleosome occupancies. Data for the middle quintile for RNAPII occupancy are shown in Fig. 3e, with the remaining quintiles shown in Supplementary Figs. 9 and 10. As expected, nucleosome-favoring sequences had increased nucleosome signals, as determined using MNase-seq. Also, despite similar RNAPII occupancies, regions with increased nucleosome occupancies showed enhanced NET-seq, chromatin RNA-seq, and CRAC-seq signals (Fig. 3e and Supplementary Fig. S9), consistent with slower RNAPII passage through these nucleosomes. When normalizing for differing nucleosome occupancy [$\log_2$(ChIP/input)], intrinsically stable nucleosomes were enriched for H4K12ac, but not Epl1, both under steady-state and TSA-treated conditions (Fig. 3e, Supplementary Figs. 9 and 10). Similar results were observed with H4K8ac and H3K23ac (Fig. 3e, Supplementary Figs. 9 and 10). Collectively, this work supports a model in which transcription-disrupted nucleosomes are acetylated by available HATs, targeted by either activators or RNAPII.

## Discussion

Previous works suggest that histone acetylation can be both a cause and consequence of transcription and in this study, we sought to define the relative contribution of these pathways to total acetylation. We found that the majority of histone acetylation is dependent on transcription and is targeted to nucleosomes at sites of RNAPII accumulation. To understand the mechanism for targeting this PTM to active genes, we mapped occupancy of Epl1, a component of the NuA4 and Piccolo HAT complexes, in both transcribing and transcription-inhibited cells. The results show that NuA4 is targeted to promoters upstream of transcription, while Piccolo binds gene bodies in a transcription-dependent manner. At this time, the mechanism for targeting Piccolo to gene bodies is unclear. Although Piccolo contains an H3K4 methyl-reader domain, we found that H3K4 methylation was dispensable for transcription-dependent acetylation[27]. This is reminiscent of previously published work showing that the Yng2 PHD finger is unnecessary for Epl1 targeting[27]. In addition, while *ESA1* is required for transcription of approximately half of all yeast genes[49], *set1Δ* mutants have defects in expression of only a small number of genes[13], and the role played by H3K4 methylation in gene activation is unclear[50]. Together, these data suggest that another pathway must exist for recruiting Piccolo to transcribed genes. The mammalian homolog of Esa1, Tip60, is proposed to be targeted via R-loops[51], which has not been reported for NuA4. However, as Esa1 contains a nucleic acid binding domain[52,53], this may be a conserved mechanism of HAT recruitment to transcribed regions.

Although histone acetylation requires the interaction of HATs with chromatin, genome-wide HAT occupancy is a poor predictor of this PTM, suggesting additional modes of regulation. In vitro and in silico studies suggest that the histone tails are not exposed to the solution, but rather interact strongly with nucleosomal DNA[43–46], which could make them poor substrates for HATs[54]. However, when transcribing through a nucleosome,

RNAPII pauses at multiple positions[55–59], inducing the formation of partially unwrapped intermediates that likely disrupt the interaction of histone tails with DNA. As such, the accumulation of RNAPII at 5′ genic regions, either due to slow passage or premature termination[60], increases the opportunity for available HATs to access the histone tails resulting in increased acetylation in these regions. Consistent with this model, we found increased acetylation on nucleosomes predicted to impede RNAPII passage. Histone acetylation is proposed to facilitate transcription by directly modulating histone–DNA contacts or by targeting chromatin remodelers to disrupt nucleosomes[61]. Thus, our research suggests that acetylation is a component of a feed-forward loop that maintains the expression of active genes.

## Methods

**Cell Culture.** FUCCI reporter mESCs[62] were grown in standard feeder-free conditions in complete mESC media: Dulbecco Modified Eagle's Medium–high glucose, 15% fetal bovine serum (HyClone Laboratories), 20 mM HEPES, 1 mM L-glutamine, 100 U/ml penicillin–streptomycin, 1 mM nonessential amino acids, ~10–50 ng/ml of recombinant LIF, 1 mM sodium pyruvate, and 0.1 mM β-mercaptoethanol on 0.2% type A gelatinized tissue culture plates.

**Yeast strains and growth.** All strains used in this study were isogenic to S288C and are listed in Supplementary Table 1. Yeast culture and genetic manipulations were performed using standard protocols. Genomic deletions were verified by polymerase chain reaction (PCR) analysis and whole-cell extracts were generated as previously described[63].

**Drug treatments.** Yeast drug treatments were performed in YPD media at the following concentrations: 400 μg/ml 1,10 phenanthroline monohydrate (Sigma 161-0158, dissolved in ethanol), 10 μg/ml thiolutin (Santa Cruz SC-200387, dissolved in DMSO), 1 mM 1-naphthalene acetic acid (Sigma N0640, dissolved in 85% ethanol), 20 μg/ml doxycycline (Sigma D9891, dissolved in 50% ethanol), 5 μM 1-Naphthyl PP1 (Sigma CAS 221243-82-9, dissolved in DMSO), 25 μM trichostatin A (dissolved in DMSO), 10 μM α factor (Sigma custom synthesized peptide, WHWLQLKPGQPMY, dissolved in 100 mM sodium acetate, pH=5.2). mESCs were treated with Actinomycin D at 25 μg/ml (Sigma CAS 50-76-0, dissolved in DMSO).

**Immunoblot analysis.** Whole-cell lysates or cellular fractions were analyzed by SDS-PAGE using the antibodies listed in the key resource table (Supplementary Table 2) at 1/1000 dilutions. Blots were scanned and fluorescent signal quantified using the Licor Odyssey scanner.

**ChIP-seq.** Yeast cells, grown to mid-log phase, were arrested in G1 by 3-h treatment with 10 μM alpha factor. Cell synchronization was verified by cell "shmooing," as seen under the microscope. For transcription inhibition, cells were treated with 400 μg/ml 1,10 phenanthroline monohydrate or 25 μM TSA for 15 m. Cells were crosslinked in 1% formaldehyde for 15 m and quenched with the addition of liquid glycine to 125 mM for a further 15 m. Cells were lysed by bead beating, and cell lysate was spun down at 15,000g for 30 m.

For sonicated ChIP-seq, the pellet was resuspended in lysis buffer (50 mM HEPES, pH 7.5, 140 mM NaCl, 0.5 mM EDTA, 1% Triton X-100, 0.1% sodium deoxycholate) and sonicated (Bioruptor, Diagenode) to produce an average fragment size of 250 bp. The lysate was spun down at 9000g for 10 m, and the supernatant was precleared by rotating with Protein G Dynabeads for 1 h at 4 °C. Twenty percent of the lysate was reserved for input, and the remaining was incubated with α-HA antibodies overnight at 4 °C.

For MNase ChIP-seq, the pellet was resuspended in MNase digestion buffer (0.5 mM spermidine, 1 mM β-ME, 0.075% NP-40, 50 mM NaCl, 10 mM Tris pH 7.4, 5 mM MgCl₂, 1 mM CaCl₂). Samples were incubated with 100 units of MNase for 10 m at 37 °C. Lysates were clarified by centrifugation at 9000g for 10 m. To extract insoluble chromatin, pellets were resuspended in 200 μl of lysis buffer with 0.2% SDS and sonicated in a Diagenode Bioruptor at the medium output for 30 s on and 30 s off for four cycles, before centrifugation at 9000g for 10 m. The second supernatant was pooled with the first, and the buffer composition of the lysate was adjusted to that of the original lysis buffer (50 mM HEPES pH 7.5, 140 mM NaCl, 1 mM EDTA, 2% Triton X-100, 0.2% Na-deoxycholate, 1× Roche protease inhibitor cocktail, 1 mM PMSF). The supernatant was precleared by rotating with Protein G Dynabeads for 1 h at 4 °C, 10% of the lysate was reserved for input, and immunoprecipitations were performed using α-HA, α-H3K23ac, α-H4K12ac, or α-H4K8ac antibodies.

Antibody immunoprecipitations were isolated by adding magnetic Protein G Dynabeads and rotating at 4 °C for 1 h, and 5 minute washes were performed twice with lysis buffer, twice with high salt buffer (50 mM HEPES pH 7.5, 640 mM NaCl, 1 mM EDTA, 2% Triton X-100, 0.2% Na-deoxycholate), twice with LiCl wash

buffer (10 mM Tris-HCl pH 8.0, 250 mM LiCl, 0.6% NP-40, 0.5% Na-deoxycholate, 1 mM EDTA), and once with TE. Synthetic spike-in DNA was added to eluates, to aid in quantification. Following proteinase K digestion, DNA was purified by phenol, chloroform, isoamyl alcohol extraction and RNase A treated.

**ChIP-seq library preparation.** Libraries for paired-end sequencing were constructed using a custom procedure for paired-end sequencing[64]. Briefly, 2–10 ng of ChIP material was end-repaired and A-tailed before being ligated to TruSeq PE adaptors. The adaptor-ligated material was subject to 8–11 rounds of PCR amplification, and an aliquot of each library was run on an Agilent Tape Station to check the size distribution and molarity of the PCR products. Equimolar amounts of indexed, amplified libraries were pooled, and fragments in the 200–600 bp size range were selected on an agarose gel. An aliquot (1 μl) of the library pool was run on an Agilent Tape Station to confirm proper size selection. In between each reaction, the material was purified using NucleoMag solid-phase reversible immobilization paramagnetic beads.

**Analysis of ChIP-seq data.** Adapter sequences were removed from paired-end FASTQ files using cutadapt (version 1.83 – http://cutadapt.readthedocs.io/en/stable/), before aligning to the saccer3 genome using BWA (version 0.7.15-r1140)[65]. For analysis of $epl1_{(1-485)}$ data, reads of chrXII were removed from all data sets, as this chromosome appeared to be unstable in this mutant (1.5× coverage). Coverage tracks represent reads per genome coverage, calculated using the Java Genomics Toolkit (https://github.com/timpalpant/java-genomics-toolkit) scripts, ngs.BaseAlignCounts and wigmath.Scale. Log₂ transformed ChIP over input tracks were calculated using the Java Genomics Toolkit and regions without signal in the input were removed to avoid division by 0. Replicates were pooled for subsequent analysis, and figures were generated in R. FASTQ files from Weiner et al. (2015) were mapped to the saccer3 genome using BWA version 0.7.15-r1140[65]. Reads were extended to 146 bp and read per genome coverage and log₂ transformed ChIP over input files were calculated using deepTools version 3.02[66]. FASTQ files from Steunou et al. (2016) were similarly mapped, but extended to 350 bp.

Similar to other groups[26,27], ChIP-seq datasets from 1,10-pt-treated or $Epl1_{(1-485)}$ cells were normalized to silent regions. The genome was divided into 250 bp bins, bins outside the interquartile range for coverage in the input were discarded, the 100 regions with the lowest Rpb3 signal were defined as silent regions, and these silent regions were used to normalized ChIP-seq datasets for cross-condition comparisons (Supplementary Table 3). We also added synthetic DNA spike-ins to our ChIP eluates and inputs (Supplementary Table 4), but this approach to normalization did not work well for all samples, possibly due to the low coverage of the spike-ins in some samples.

**Defining genome annotations.** Yeast transcription start and end sites were downloaded from the supplemental files of Chereji et al.[67]. To identify active, non-divergent, yeast promoters, genes in the lowest quintile of NET-seq signal over the first 500 bp downstream of the TSS were designated as non-transcribed. Uni-directional promoters were then defined as transcribed genes with the lowest quintile of NET-seq signal 100–600 bp upstream of the TSS (832 genes). RefSeq mm9 TSSs were downloaded from the UCSC Genome Browser (https://genome.ucsc.edu/). To identify mouse genes with active, non-divergent, promoters, transcribed genes were defined as those with greater than the median PRO-seq signal over 1 kb downstream of the TSS, and transcribed genes in the lowest quintile of PRO-seq signal upstream of the TSS were designated as having unidirectional promoters (3035 genes).

For transcribed nucleosomes classified by Rpb3 change upon 1,10-pt treatment (Supplementary Figs. 3B and 5B), genome-wide nucleosome positions[47] with Rpb3 signal greater than the median were classified as transcribed. Nucleosomes, where Rpb3 changed by less than 10%, were classified as "Rpb3 stable", while those decreasing by at least 3× were classified as "Rpb3 lost". Boxplots represent the first to third quartiles, with whiskers extending to 1.5 times the interquartile range or to the extreme of the data. Notches are equal to ±1/58 IQR /sqrt($n$), giving an approximation of the 95% confidence interval for the difference in 2 medians.

To find promoter peaks of Epl1, the Epl1 MNase ChIP-seq was compared to its input within the NDR for each gene[67]. Within each NDR, a smoothing spline was fit to the IP minus input signal (RPGC) and the peak position was selected. Peak positions with an IP minus input greater than 0.5 RPGC in the Epl1 ChIP-seq but not in the untagged control were selected as Epl1 peaks. NDRs in close proximity to tRNA genes or centromeres were removed from further analysis due to the binding of Epl1 to these elements.

For motif analysis, Epl1 peaks were compared to 1958 NDR regions depleted for Epl1 binding (maximum IP minus input less than 0.1 RPGC). The 500 bp regions around peak centers were then inputted into the MEME-ChIP Differential Enrichment algorithm[68] to find enriched motifs from the JASPAR nonredundant core fungi motifs[69]. For the top two hits, Rap1 and Aft2, CentriMo[70] was used to plot the distance from the best motif site to the Epl1 peak center and the motif probabilities around the best motif site for the regions containing target motifs.

**Generating heatmaps and metaplots.** Metaplot matrices centered on TSSs were generated using the sitepro script from the CEAS package 1.0.2[71] and matrices

aligned to other features were produced using the visualization.MatrixAligner script from the Java Genomics Toolkit. Heatmaps were generated using deep-Tools[66], and for 2D heatmaps, plot2DO (version 1) was used[72].

**Reporting summary**. Further information on research design is available in the Nature Research Reporting Summary linked to this article.

## Data availability

Data generated for this manuscript were deposited in the NCBI Gene Expression Omnibus under the accession code "GSE110287". Published datasets analyzed for this paper include "SRP132377" (*S. cerevisiae* RNAPII ChIP-seq ± 1,10-pt[73]), "SRP095935" (*S. cerevisiae* H3K23ac ± TSA and Sas3 ChIP-seq[33]), "SRP048526" (*S. cerevisiae* histone PTM MNase ChIP-seq[47]), "GSE68484" (*S. cerevisiae* NET-seq[41]), "GSE68484" (*S. cerevisiae* chromatin RNA-seq[41]), "GSE69676" (*S. cerevisiae* CRAC-seq[40]), http://sgd-archive.yeastgenome.org/published_datasets/MacIsaac_2006_PMID_16522208/track_files/MacIsaac_2006_ChIP_chip_TFBSs_V64.gff3 (*S. cerevisiae* TFBS[38]), "SRP070154" (*S. cerevisiae* Epl1 and H4K8ac ChIP-chip[27]), "GSE36600" (*S. cerevisiae* Gcn5 and H3K18ac ChIP-chip[32]), http://sgd-archive.yeastgenome.org/published_datasets/Kaplan_2009_PMID_19092803/track_files/Kaplan_2009_predicted_average_nucleosome_occupancy_V64.wig (*S. cerevisiae* Predicted nucleosome occupancy[48]), https://www.ncbi.nlm.nih.gov/pmc/articles/PMC5807854/bin/13059_2018_1398_MOESM2_ESM.xlsx (*S. cerevisiae* TSS and PAS annotations[67]), https://www.ncbi.nlm.nih.gov/pmc/articles/PMC4405355/bin/mmc3.csv (*S. cerevisiae* genome-wide nucleosome positions[47]), "GSE130691" (*M. musculus* mESC PRO-seq[74]), "GSE31039" (*M. musculus* mESC H3K9ac and H3K27ac ChIP-seq, Mouse ENCODE epigenomic data). All relevant data supporting the key findings of this study are available within the article and its Supplementary Information files or from the corresponding author upon reasonable request. Source data for Fig. 1a, d, and Epl1 peak midpoints are provided in the Source Data file. A reporting summary for this Article is available as a Supplementary Information file. Source data are provided with this paper.

## Code availability

The following software packages were used in this study: cutadapt (version 1.83—http://cutadapt.readthedocs.io/en/stable/), BWA (version 0.7.15-r1140)[65], Java Genomics Toolkit (https://github.com/timpalpant/java-genomics-toolkit), deepTools version 3.02[66], CEAS package version 1.0.2[71], plot2DO[72], the MEME-ChIP Differential Enrichment algorithm[68], and CentriMo[70].

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

## Acknowledgements

Support for this work was provided by grants to L.J.H. and M.C.L. from the Canadian Institutes of Health Research (PJT-162253) and Natural Sciences and Engineering Research Council (RGPIN-2018-04907). B.J.E.M. was supported by a fellowship from the Natural Sciences and Engineering Research Council. We are grateful to Steven Hahn, Michael Kobor, and Hiroshi Kimura for providing plasmids and antibodies.

## Author contributions

Conception: L.J.H. and B.J.E.M.; acquisition: B.J.E.M., J.B., A.K., K.N., and J.L.; analysis: L.J.H. and B.J.E.M.; drafting, B.J.E.M.; revision, B.J.E.M., L.J.H., J.B., and M.C.L.

## Competing interests

The authors declare no competing interests.
