## [Peer Review File · Nature Communications]

Reviewers' comments:

Reviewer #1 (Remarks to the Author):

The prevailing model is that histone acetylations are targeted to promoter regions by interactions between activators and HATs. These acetylations in turn help recruit bromodomain proteins found in ATP-dependent remodelers and TFIID (as well as the HATs themselves) to open up NFRs and allow PIC assembly. This is an interesting paper presenting data that the majority of acetylation is temporally downstream, not upstream, of transcription. The bulk western blots are convincing, showing acetylation drops upon several different treatments that inhibit transcription. There have been multiple papers suggesting that the SAGA and NuA4 HATs can be targeted to transcribed regions, yet experiments here argue against NuA4 (as monitored via its Epl1 subunit) targeting being due to CTD phosphorylation or H3K4 methylation as previously proposed. The authors then come to the surprising conclusion that acetylation does not even correlate with the presence of the HATs, which leads them to propose that something occurs during the process of transcription to directly or indirectly stimulate acetylation. Overall, it's a pretty radical paper. I'm convinced that transcription somehow stimulates overall acetylation levels, but some of the later conclusions seem weak or speculative to me. Below are some specific comments and questions that could clear up my doubts.

1. If I understand correctly, the paper suggests that HATs can be recruited to transcribed regions independently of activators (or CTD or histone methylation), and HATs can be recruited to promoters by activators, but in all cases acetylation requires ongoing transcription. The bulk chromatin westerns assay both. From sums of the ChIP reads, can the authors determine whether the western signals are dominated by promoter regions (high levels in short peak regions) or gene bodies (low levels but much larger areas of the genome)? This would help (me at least) think about how to reconcile the western and ChIP data.
2. In Figs 3 and S5, I'm confused about the disconnect between the predicted nucleosome occupancy, which shows large effects; the MNase data, where there is little difference between the high/low classes; and the histone modifications, where the differences seem substantial. All three of these graphs use different metrics on the y-axes, but the paper conclusions hinge on making quantitative comparisons between them. At the very least, the histone modifications (3G and H) should be normalized to total histone.
3. Fig 3B. The Spearman correlation coefficient depends on the co-linearity of both datasets. However, it's not clear that all these datasets have linear signals throughout their entire dynamic range. The correlations appear to be highly dependent on the particular data set (for example, NET-seq and PRO-seq should be highly correlated with each other, yet show very different coefficients with acetylations), so I wonder if Pearson correlations (which test for correlation of order without requiring strictly co-linearity) would be more revealing.
4. Fig S4. Do these graphs cover the entire genome? Or only Epl1 binding sites? I'm trying to understand if there is no correlation of acetylation levels with the magnitude of Epl1 at called binding site peaks (which I can buy), or that there is no correlation of acetylation with the presence or absence of Epl1 (which would be harder to explain, since you presumably can't get acetylation if the HAT is never there).

Minor point:

Fig 2A - missing an explanation of the gray line in the key (looks like input?)

Reviewer #2 (Remarks to the Author):

In this article, Martin et al have focused on the effect of inhibition of transcription by 1,10-pt on the acetylation of some of the histone H3 and H4 lysines genome-wide using *S. cerevisiae* as a model system. Authors also performed immunoblot to measure the global loss of acetylated histones in mouse ES cells upon treatment with transcriptional inhibitors. Based on ChIPseq for they infer that transcription promotes interaction of lysine acetyltransferases (KAT/HAT) – Epl1 to chromatin. Furthermore, they suggest histone acetylation is regulated by RNA Polymerase II activity after recruitment of KATs to chromatin. Overall, I find this study interesting and important contribution to the transcription and chromatin field. However, I find their experiments and results are not sufficient to conclude that the majority of histone acetylation is a consequence of transcription.

1) Especially the mechanism of action of 1,10pt on transcription is not clear. Can authors rule out the possibility of this drug not affecting the recruitment/activity of acetyltransferases? Other better-studied transcription inhibitors like DRB which are known to affect transcriptional elongation will provide insights into transcription-dependent effect on acetylation.

2) The title is too broad and it does not justify by the experiments and the approaches used in this study.

3) Histone acetylations like H4K16ac, H4K64ac, H3K56ac and H3K122ac are previously shown to have a causal role in transcription both in vitro assays and mutational studies in yeast. Possibility of role of some of the acetylations in transcription and chromatin structure is not mentioned in this study. Why did the authors exclude these modifications, it will be good to include some of these acetylations in this study.

4) Fig 1C shows immunoblots are done for H3K27ac and H3K9ac in mESC to show global loss of acetylations upon Actinomycin D treatment, and in the results section authors concluded that in yeast and mammalian system acetylation is dependent on transcription. What happens to other acetylations like H4K16ac, H3K36ac, H3K122ac, H3K56ac and H3K64ac in mESCs. It is important to probe some of these acetylations in mESCs.

5) Although immunoblots show reduced acetylation levels in mESCs it is possible that acetylation is still maintained at regulatory elements and genes, ChIPseq for acetylations e.g. H3k27ac and H4K16ac in mESCs upon transcription inhibition will strengthen the conclusion.

6) Authors need to discuss some of the earlier observations for e.g.

a) dCas9 mediated recruitment of wild type but not catalytically inactive KAT (p300) to enhancers and promoters is known to activate transcription. doi: 10.1038/nbt.3199.

b) Acetylation precedes transcription in developmental genes doi: 10.1242/dev.179127

7) Although the author's correlation analysis shows histone methyltransferase level correlate with gene expression. Several pieces of evidence show H3K4 methylation also may not have an instructive role in transcription. It will be good to add this in the discussion. DOI

10.1002/bies.201600095

Reviewer #3 (Remarks to the Author):

Martin et al. describe results from a number of experiments designed to understand the extent of causality between histone acetylation and transcription, predominantly in *S. cerevisiae*, with additional data from murine embryonic stem cells. Based on a series of genomic experiments

carried out in the presence or absence of transcription inhibitors, the authors conclude that histone acetylation occurs largely as a consequence of RNA Pol II transcription, rather than the leading current model of acetylation as a precursor to transcription. The authors note that while HATs are depleted from gene bodies, HAT occupancy and targeting alone are insufficient to explain the changes seen upon inhibition of transcription, suggesting instead that HAT activity is regulated post-recruitment in a Pol II transcription-dependent manner.

The authors have presented thorough and compelling data that is well-suited for Nature Communications; however, I have the following concerns:

Major Comments:

1. Can the authors disentangle transcription initiation and elongation as they relate to subsequent histone acetylation? Actinomycin D is a transcription elongation inhibitor, while thiolutin (and likely 1,10-pt) are inhibitors of transcription initiation; however, elongation was only inhibited in mESCs, and initiation was inhibited in yeast. Do the authors have targeted inhibition of each stage in one system that can address this question?

2. The authors conclude from Epl1 ChIPseq data presented in Figure 2 that RNAPII is required to target HATs. To strengthen the conclusions drawn from these data, additional analyses regarding the data presented in Figure 2B could help, as has been done very nicely in Figure 1E. For example, perhaps breaking the data down into Epl1-lost genes, Epl1-gained genes, and Epl1-unchanged genes and then sorting by RNAPII loss would help to clarify the data. Perhaps breaking the genes into categories for divergent transcription vs unidirectional transcription as was discussed in the manuscript would lead to interesting observations. In addition, the 562 that have Epl1 peaks could be detailed more: what type of motifs are present at these genes? Do these motifs correspond to any activators that could help strengthen data shown in Figure 2C?

3. Could the authors explain why data shown in Figures 3C-H are analyzed over +2, +3, and +4 nucleosomes, when the +1 nucleosome appears to be most affected according to the data shown in Figure 1? Relatedly, do the authors have an explanation as to why the +1 nucleosome is most affected by 1,10-pt treatment in Figure 1D?

Minor Comments:

1. In general, some conclusions are slightly overstated based on the evidence provided. I encourage the authors to reexamine their data and stray away from broad conclusions that are not directly tested or shown (e.g. page 2, paragraph 1, lines 6-7: Does a correlation between histone acetylation levels and transcription imply causation?; page 6, paragraph 1, lines 5-8; Does deacetylation primarily at regions that lose Pol II upon transcription initiation indicate a direct effect? Or may a transcription-dependent indirect effect still be at play?).

2. The data presented in Figure 1 are very compelling; however, they lack an important control, which is making the analysis relative to total histone levels. Perhaps including ChIPseq for total H3 and/or H4 or MNase-seq upon 1,10-pt treatment.

3. Inclusion of blots from mESCs to strengthen the conservation across systems is nice. However, the authors only examine two histone acetylation marks. The same antibodies used for the yeast protein extracts should work just fine to probe more modifications from mESC protein extracts.

4. Include quantitation of RNA Pol II levels in Figure 1A alongside the Ser5P blots.

5. In Figure 2A and 2C, plot these comparable analyses with the same x axis to avoid confusion.

6. In Figure 3A, the y-axis should be labeled to include both H4K12ac and Epl1 data, as both are shown in the graph.

7. Do the CRAC-seq and chromatin RNA-seq show the same trend as the NET-seq data shown in Figure 3D? It appears the authors analyzed these data and could include in the supplement to strengthen the conclusions.

8. In Figure 3E, is this the Rpb3 ChIP-exo (Van Oss et al) or RNAPII ChIP-seq (Schultz et al). I believe it is the ChIP-exo, however it is labeled RNAPII ChIP-exo, whereas in Figure 3B they are

listed as Rpb3 ChIP-exo and RNAPII ChIP-seq. Maintaining the same reference language would help readers.

9. In Figure S1D and S1E, change "Mock" to "DMSO" to more explicitly state the experimental conditions.

10. Why are the raw Western blots for Figure S3B are not shown, when all other related figures throughout the paper include the accompanying blots?

11. I did not see reference to Figure S5 in the body of the text. Also, why are a different number of genes analyzed in Figure S5 when compared with Figure 3F-H; if TSA treatment led to different nucleosome occupancy and positioning, doesn't that warrant some discussion? Additionally, the legend is inconsistent with Figure 3.

12. I have identified numerous typos throughout the manuscript (e.g. "Elp1" on page 7, paragraph 1, line 10 and page 8, paragraph 2, line 1; "approach allowed is to identify" on page 7, paragraph 1, line 4; "RNAPII ChIP-seq signal" on page 10, paragraph 1, line 9, when the experiment referred to is actually ChIP-exo data; Figure S5 legend, line 6: "a Kaplan score greater and 0.5"). I encourage the authors to carefully reexamine their manuscript to ensure that their message remains clear and correct.

All my best,
Sarah Hainer

Reviewer #1 (Remarks to the Author):

The prevailing model is that histone acetylations are targeted to promoter regions by interactions between activators and HATs. These acetylations in turn help recruit bromodomain proteins found in ATP-dependent remodelers and TFIID (as well as the HATs themselves) to open up NFRs and allow PIC assembly. This is an interesting paper presenting data that the majority of acetylation is temporally downstream, not upstream, of transcription. The bulk western blots are convincing, showing acetylation drops upon several different treatments that inhibit transcription. There have been multiple papers suggesting that the SAGA and NuA4 HATs can be targeted to transcribed regions, yet experiments here argue against NuA4 (as monitored via its Epl1 subunit) targeting being due to CTD phosphorylation or H3K4 methylation as previously proposed. The authors then come to the surprising conclusion that acetylation does not even correlate with the presence of the HATs, which leads them to propose that something occurs during the process of transcription to directly or indirectly stimulate acetylation. Overall, it's a pretty radical paper. I'm convinced that transcription somehow stimulates overall acetylation levels, but some of the later conclusions seem weak or speculative to me. Below are some specific comments and questions that could clear up my doubts.

1. If I understand correctly, the paper suggests that HATs can be recruited to transcribed regions independently of activators (or CTD or histone methylation), and HATs can be recruited to promoters by activators, but in all cases acetylation requires ongoing transcription. The bulk chromatin westerns assay both. From sums of the ChIP reads, can the authors determine whether the western signals are dominated by promoter regions (high levels in short peak regions) or gene bodies (low levels but much larger areas of the genome)? This would help (me at least) think about how to reconcile the western and ChIP data.

If we define “promoter regions” as just the +1 and -1 nucleosomes, then 22% of H3K23ac and 18% of H4K8ac/H4K12ac ChIP-seq reads map to promoters and ~80% map elsewhere. If we expand promoter regions to include +1, +2, -1, and -2 nucleosomes, then 40% of H3K23ac and 35% of H4K8ac/H4K12ac is found at promoters. Thus, a large portion of histone acetylation is not immediately adjacent to promoters.

2. In Figs 3 and S5, I'm confused about the disconnect between the predicted nucleosome occupancy, which shows large effects; the MNase data, where there is little difference between the high/low classes; and the histone modifications, where the differences seem substantial. All three of these graphs use different metrics on the y-axes, but the paper conclusions hinge on making quantitative comparisons between them. At the very least, the histone modifications (3G and H) should be normalized to total histone.

In the previous version of this manuscript, histone acetylation in Figures 3 and 5 was normalized to total nucleosomes, and we apologize for not making this clear. We also apologize for designing an overly complicated experiment and would like to thank this reviewer for making us rethink how best to perform this analysis. In this figure, we initially sought to identify nucleosomes that were strong barriers to transcription to determine whether nucleosomes that impede RNAPII show increased acetylation. We used the “Kaplan” score to identify nucleosomes with different predicted occupancies and then selected those with similar actual

occupancies. The hope here was that we would identify a subset of nucleosomes that were similar, except for their stability. However, as pointed out by this reviewer, the correlation between predicted and actual nucleosome occupancies left us few nucleosomes that differ in these parameters to analyze. We now realize that, because we normalize acetylation levels to total nucleosomes, it isn't necessary to control for nucleosome occupancy. In the new analysis (Figures 3E, Supplementary Figures 9 and 10 in the revised manuscript), we divided genic nucleosomes first into bins based on RNAPII occupancy, and then into bins based on predicted nucleosome occupancy. This allowed us to identify pools of nucleosomes with similar levels of RNAPII but differing predicted stability. As expected, nucleosomes within each bin differed in actual occupancy, but when normalized for this [$\log_2(\text{ChIP}/\text{input})$], histone acetylation levels increased with predicted nucleosome stability. Another advantage of this approach is that it gave us thousands, as opposed to hundreds, of nucleosomes in each bin to analyze, generating a more robust result. Another change to the revised manuscript is that we now show data for all genic nucleosomes. Figure 3E shows nucleosomes for the middle quintile of RNAPII, with the remaining quintiles shown in Supplementary Figures 9 and 10.

3. Fig 3B. The Spearman correlation coefficient depends on the co-linearity of both datasets. However, it's not clear that all these datasets have linear signals throughout their entire dynamic range. The correlations appear to be highly dependent on the particular data set (for example, NET-seq and PRO-seq should be highly correlated with each other, yet show very different coefficients with acetylations), so I wonder if Pearson correlations (which test for correlation of order without requiring strictly co-linearity) would be more revealing.

As suggested, we examined the Pearson correlations between NET-seq, CRAC-seq, and chromatin RNA-seq and acetylation and found the coefficients to be less than 0.1. Part of the explanation for this is that, as this reviewer suggests, histone acetylation is not linear throughout the dynamic range. Very highly expressed genes do not show commensurate levels of histone acetylation. Excluding these genes increased the Pearson correlation coefficients to 0.2 – 0.35, with only a modest increase in Spearman coefficients. We believe this effect is due to the high level of histone turnover that occurs on highly expressed genes (PMID: 17347438), but without confirmation of this, we have decided to remove the correlation analyses from the revised manuscript.

4. Fig S4. Do these graphs cover the entire genome? Or only Epl1 binding sites? I'm trying to understand if there is no correlation of acetylation levels with the magnitude of Epl1 at called binding site peaks (which I can buy), or that there is no correlation of acetylation with the presence or absence of Epl1 (which would be harder to explain, since you presumably can't get acetylation if the HAT is never there).

Again, these figures have been removed from the revised manuscript. To address this reviewer's question, these graphs did cover the entire genome. We agree that regions without HATs should not be acetylated, however consistent with the work of others (PMID: 27550811), it seems that most regions contain some Epl1.

Minor point:

Fig 2A - missing an explanation of the gray line in the key (looks like input?)

Because yeast genes tend to be short, in our metagene plots we ignore data downstream of polyadenylation sites, and the grey line represents the fractions of genes that are plotted. We apologize for this omission and this is now described in all relevant figure legends.

Reviewer #2 (Remarks to the Author):

In this article, Martin et al have focused on the effect of inhibition of transcription by 1,10-pt on the acetylation of some of the histone H3 and H4 lysines genome-wide using *S. cerevisiae* as a model system. Authors also performed immunoblot to measure the global loss of acetylated histones in mouse ES cells upon treatment with transcriptional inhibitors. Based on ChIPseq for they infer that transcription promotes interaction of lysine acetyltransferases (KAT/HAT) – Epl1 to chromatin. Furthermore, they suggest histone acetylation is regulated by RNA Polymerase II activity after recruitment of KATs to chromatin. Overall, I find this study interesting and important contribution to the transcription and chromatin field. However, I find their experiments and results are not sufficient to conclude that the majority of histone acetylation is a consequence of transcription.

1) Especially the mechanism of action of 1,10pt on transcription is not clear. Can authors rule out the possibility of this drug not affecting the recruitment/activity of acetyltransferases? Other better-studied transcription inhibitors like DRB which are known to affect transcriptional elongation will provide insights into transcription-dependent effect on acetylation.

One of the drawbacks of using yeast is that they are not permeable to many drugs, including actinomycin D, triptolide, and DRB. As described in previous work (PMID: 20890900), the addition of polygodial to permeabilize yeast cell membranes can improve drug uptake, but we found that this reagent had little impact on the sensitivity of yeast to these drugs. We therefore used 1,10-phenanthroline and thiolutin to chemically inhibit transcription, and an auxin-inducible degron on Rpb2 to genetically ablate RNAPII. We argue that loss of acetylation in 1,10-pt is not due to reduced HAT targeting, as acetylation is lost despite unchanged (Figures 2B and G) or increased (Figures 2E and F) HAT occupancy at promoters. Finally, the tight correspondence between where RNAPII and acetylation are lost upon drug treatment (Supplementary Figure 3) argues that we are not globally inhibiting HAT activity as acetylation is still present at regions that do not lose RNAPII.

2) The title is too broad and it does not justify by the experiments and the approaches used in this study.

We agree with this critique and have changed the title to “Transcription shapes genome-wide histone acetylation patterns”, which we feel better reflects the conclusions of this study.

3) Histone acetylations like H4K16ac, H4K64ac, H3K56ac and H3K122ac are previously shown to have a causal role in transcription both in vitro assays and mutational studies in yeast. Possibility of role of some of the acetylations in transcription and chromatin structure is not

mentioned in this study. Why did the authors exclude these modifications, it will be good to include some of these acetylations in this study.

In the revised manuscript we have added immunoblot data for H4K16ac, H3K56ac, and H3K122ac (Figure 1C). Interestingly, we did not see a signal with the H3K64ac antibody, which could suggest lack of antibody specificity or that acetylation of H3K64 is mammalian-specific.

4) Fig 1C shows immunoblots are done for H3K27ac and H3K9ac in mESC to show global loss of acetylations upon Actinomycin D treatment, and in the results section authors concluded that in yeast and mammalian system acetylation is dependent on transcription. What happens to other acetylations like H4K16ac, H3K36ac, H3K122ac, H3K56ac and H3K64ac in mESCs. It is important to probe some of these acetylations in mESCs.

Unfortunately, due to the reasons outlined in our cover letter, we were unable to perform additional experiments with mammalian cells. Instead, we have added analysis of published mESC data to demonstrate that acetylation upstream of promoters is associated with divergent transcription (Figure 1H).

5) Although immunoblots show reduced acetylation levels in mESCs it is possible that acetylation is still maintained at regulatory elements and genes, ChIPseq for acetylations e.g. H3k27ac and H4K16ac in mESCs upon transcription inhibition will strengthen the conclusion.

We agree that additional work with mESCs could yield some interesting insights. Most mammalian HATs have homologs in yeast, with the exception of p300/CBP, which colocalize with H3K27ac at regulatory elements. Thus, an intriguing hypothesis is that p300/CBP acetylate histones upstream of transcription. Contrary to this however, Bose et al., 2017 (PMID: 28086087) showed that the stimulation of CBP acetyltransferase activity by enhancer RNAs results in transcription-dependent changes in histone acetylation. Thus, although we believe the underlying mechanism to be different, evidence suggests that acetylation by p300/CBP is also transcription-dependent.

6) Authors need to discuss some of the earlier observations for e.g.

a) dCas9 mediated recruitment of wild type but not catalytically inactive KAT (p300) to enhancers and promoters is known to activate transcription. doi: 10.1038/nbt.3199.

In addition to histones, acetylation of over a hundred non-histone, transcription-regulating proteins has been reported, including transcription factors, transcriptional co-activators and nuclear receptors (PMID: 30467427). As such, even if all histone acetylation was a consequence of transcription (which we are not arguing is the case), it would not be surprising if “HATs” were still required for gene activation. We mention the potential contribution of non-histone acetylation in the Introduction and Discussion.

b) Acetylation precedes transcription in developmental genes doi: 10.1242/dev.179127

This study applied Fab-based single cell imaging to monitor the changes in histone PTM levels during zygotic genome activation (ZGA) in zebrafish embryos. The key result was that treatment

of cells with α -amanitin prevented formation of s2p but not H3K27ac, which seems to directly contradict our results. There are several possible explanations for this discrepancy. First, as mentioned above, yeast lack p300/CBP, the HAT responsible for H3K27ac in metazoans. Oddly, in the Sato et al study, attempts to inhibit p300 failed to reduce H3K27ac levels suggesting either the Fabs used were not specific to H3K27ac or there is a novel H3K27ac-specific HAT in zebrafish. Second, the results of our study suggest that histone acetylation is dependent on nucleosome disruption by RNAPII. It is possible that during ZGA, an alternate force disrupts chromatin, facilitating acetylation. One possibility is chromatin remodellers.

7) Although the author's correlation analysis shows histone methyltransferase level correlate with gene expression. Several pieces of evidence show H3K4 methylation also may not have an instructive role in transcription. It will be good to add this in the discussion. DOI 10.1002/bies.201600095

We thank this reviewer for this suggestion and have referenced this information in the first paragraph of the Discussion.

Reviewer #3 (Remarks to the Author):

Martin et al. describe results from a number of experiments designed to understand the extent of causality between histone acetylation and transcription, predominantly in *S. cerevisiae*, with additional data from murine embryonic stem cells. Based on a series of genomic experiments carried out in the presence or absence of transcription inhibitors, the authors conclude that histone acetylation occurs largely as a consequence of RNA Pol II transcription, rather than the leading current model of acetylation as a precursor to transcription. The authors note that while HATs are depleted from gene bodies, HAT occupancy and targeting alone are insufficient to explain the changes seen upon inhibition of transcription, suggesting instead that HAT activity is regulated post-recruitment in a Pol II transcription-dependent manner.

The authors have presented thorough and compelling data that is well-suited for Nature Communications; however, I have the following concerns:

Major Comments:

1. Can the authors disentangle transcription initiation and elongation as they relate to subsequent histone acetylation? Actinomycin D is a transcription elongation inhibitor, while thiolutin (and likely 1,10-pt) are inhibitors of transcription initiation; however, elongation was only inhibited in mESCs, and initiation was inhibited in yeast. Do the authors have targeted inhibition of each stage in one system that can address this question?

One of the disadvantages of working with yeast is that the cells are not permeable to many drugs, including those that target specific stages of transcription (i.e. actinomycin D, triptolide, and DRB). Previous work (PMID: 20890900) used polygodial to permeabilize yeast cell membranes and improve drug uptake, but we were unable to reproduce this. Thus, we could not use the same inhibitors on yeast that are commonly used in mammalian cell culture work. We do argue,

however, that actinomycin D and 1,10-phenanthroline inhibit both transcription initiation and elongation. First, Figure 1D shows that actinomycin D causes loss of RNAPII s5p, a PTM associated with promoter escape. Second, 15 minutes of 1,10-phenanthroline treatment should have been sufficient time for polymerase clearance from most genes. The fact that there is still residual RNAPII at downstream regions (Figure 1B) suggests that 1,10-pt also impairs elongation.

2. The authors conclude from Epl1 ChIPseq data presented in Figure 2 that RNAPII is required to target HATs. To strengthen the conclusions drawn from these data, additional analyses regarding the data presented in Figure 2B could help, as has been done very nicely in Figure 1E. For example, perhaps breaking the data down into Epl1-lost genes, Epl1-gained genes, and Epl1-unchanged genes and then sorting by RNAPII loss would help to clarify the data. Perhaps breaking the genes into categories for divergent transcription vs unidirectional transcription as was discussed in the manuscript would lead to interesting observations. In addition, the 562 that have Epl1 peaks could be detailed more: what type of motifs are present at these genes? Do these motifs correspond to any activators that could help strengthen data shown in Figure 2C?

These are excellent suggestions and we have adopted all of them. A heatmap and boxplot (Supplemental Figures 5A and B) are included to support a requirement of RNAPII for targeting Epl1 to gene bodies. Figures 1A and B compare the impact of transcription inhibition at divergent vs. unidirectional promoters. Supplementary Figure 8 explores the specific motifs associated with the Epl1 peaks.

3. Could the authors explain why data shown in Figures 3C-H are analyzed over +2, +3, and +4 nucleosomes, when the +1 nucleosome appears to be most affected according to the data shown in Figure 1? Relatedly, do the authors have an explanation as to why the +1 nucleosome is most affected by 1,10-pt treatment in Figure 1D?

In our revised manuscript we altered this analysis to include all genic nucleosomes, after separating them into bins based on RNAPII occupancies (Figure 3E and Supplementary Figures 9 and 10). The results are similar to the original manuscript but now assess more of the genome. In regard to the disproportionate impact on +1 nucleosome acetylation, we argue that this is simply due to the localization of these nucleosomes in regions with the greatest loss of RNAPII as is shown in Supplementary Figure 3A.

Minor Comments:

1. In general, some conclusions are slightly overstated based on the evidence provided. I encourage the authors to reexamine their data and stray away from broad conclusions that are not directly tested or shown (e.g. page 2, paragraph 1, lines 6-7: Does a correlation between histone acetylation levels and transcription imply causation?; page 6, paragraph 1, lines 5-8; Does deacetylation primarily at regions that lose Pol II upon transcription initiation indicate a direct effect? Or may a transcription-dependent indirect effect still be at play?).

On page 2 we now state “genome-wide occupancy studies show that histone acetylation levels correlate strongly with transcription, suggesting a causal relationship between the two.” And on page 6 we now state “were primarily deacetylated at regions that lost RNAPII upon transcription

inhibition, suggestive of a direct effect”. Additionally, we have changed the title of the manuscript to “Transcription shapes genome-wide histone acetylation patterns” as this more accurately describes the results.

2. The data presented in Figure 1 are very compelling; however, they lack an important control, which is making the analysis relative to total histone levels. Perhaps including ChIPseq for total H3 and/or H4 or MNase-seq upon 1,10-pt treatment.

We have now included the MNase-seq as a separate profile in Figure 1F and all other inputs are shown in Supplementary Figure 1.

3. Inclusion of blots from mESCs to strengthen the conservation across systems is nice. However, the authors only examine two histone acetylation marks. The same antibodies used for the yeast protein extracts should work just fine to probe more modifications from mESC protein extracts.

Unfortunately, as described in our cover letter, we have been unable to generate additional data using mammalian cells. Instead, we have bolstered the mammalian data by demonstrating that acetylation upstream of promoters is associated with divergent transcription (Figure 1H).

4. Include quantitation of RNA Pol II levels in Figure 1A alongside the Ser5P blots.

We have unfortunately been unsuccessful with Rpb1 (8WG16) blots. While we realize that this is not ideal, the recovery of RNAPII s5p after only 5 minutes of 1,10-pt washout supports Rpb1 levels being unaffected by transcription inhibition (Supplementary Figure 6B).

5. In Figure 2A and 2C, plot these comparable analyses with the same x axis to avoid confusion.

We now include an additional figure (Figure 2E) in which the data in Figure 2A is plotted using the identical parameters as Figure 2D (i.e. former figure 2C).

6. In Figure 3A, the y-axis should be labeled to include both H4K12ac and Ep11 data, as both are shown in the graph.

We apologize for this error. In the revised manuscript the Y-axis is labelled as “RPGC” (reads per genomic coverage).

7. Do the CRAC-seq and chromatin RNA-seq show the same trend as the NET-seq data shown in Figure 3D? It appears the authors analyzed these data and could include in the supplement to strengthen the conclusions.

Yes, analyses of these datasets do show the same trend, which is now shown in Figure 3E and Supplementary Figure 9.

8. In Figure 3E, is this the Rpb3 ChIP-exo (Van Oss et al) or RNAPII ChIP-seq (Schultz et al). I believe it is the ChIP-exo, however it is labeled RNAPII ChIP-exo, whereas in Figure 3B they

are listed as Rpb3 ChIP-exo and RNAPII ChIP-seq. Maintaining the same reference language would help readers.

Our apologies for this error, although in response to comments from Reviewer 1, we have removed this figure from the revised manuscript.

9. In Figure S1D and S1E, change “Mock” to “DMSO” to more explicitly state the experimental conditions.

Again, our apologies for the confusion. The immunoblot data has been replotted using the term “EtOH” in lieu of “Mock”.

10. Why are the raw Western blots for Figure S3B are not shown, when all other related figures throughout the paper include the accompanying blots?

Because Nature Communications requires submission of all raw data in the supplemental, we now only include the quantification of immunoblot data in the main and supplementary figures.

11. I did not see reference to Figure S5 in the body of the text. Also, why are a different number of genes analyzed in Figure S5 when compared with Figure 3F-H; if TSA treatment led to different nucleosome occupancy and positioning, doesn't that warrant some discussion? Additionally, the legend is inconsistent with Figure 3.

Our apologies for this. As discussed above, we have changed how we performed this analysis such that we no longer include or exclude nucleosomes from the analysis based on nucleosome occupancy.

12. I have identified numerous typos throughout the manuscript (e.g. “Elp1” on page 7, paragraph 1, line 10 and page 8, paragraph 2, line 1; “approach allowed is to identify” on page 7, paragraph 1, line 4; “RNAPII ChIP-seq signal” on page 10, paragraph 1, line 9, when the experiment referred to is actually ChIP-exo data; Figure S5 legend, line 6: “a Kaplan score greater and 0.5”). I encourage the authors to carefully reexamine their manuscript to ensure that their message remains clear and correct.

Our apologies for this. All of the above-mentioned typos have been fixed. Thank you for bringing them to our attention.

All my best,
Sarah Hainer

REVIEWERS' COMMENTS

Reviewer #1 (Remarks to the Author):

I am satisfied with the responses to the points I made in my earlier review.

Reviewer #2 (Remarks to the Author):

The current version of the manuscript by Martin et al is a much-improved version and analysis is well done. Overall It is an elegant work, I commend the authors for adding new analysis and also for changing some of the discussion and conclusions of the results. Understandably, further experiments suggested in mouse ESCs are not feasible. Data shown in this manuscript clearly shows inhibition of transcription leads to reduced acetylation and transcription contributes to histone acetylation patterns. This version is suitable for publication in Nature communications.

Minor points: Authors should discuss recent publication in Genome research <http://www.genome.org/cgi/doi/10.1101/gr.257576.119>, which shows increased histone acetylation leads to increased binding of TFs and transcription. Please check grammar, e.g. Page 14 line 16

Best regards
Pradeepa Madapura

Reviewer #3 (Remarks to the Author):

The authors have done an excellent job addressing the comments from all reviewers. I find this revised manuscript to be rigorous and the conclusion well supported by the data.

October 26, 2020

We would like to thank the reviewers for their time, suggestions, and understanding. Our responses to their comments are outlined below in blue text.

Reviewer #1 (Remarks to the Author):

I am satisfied with the responses to the points I made in my earlier review.

Thank you.

Reviewer #2 (Remarks to the Author):

The current version of the manuscript by Martin et al is a much-improved version and analysis is well done. Overall It is an elegant work, I commend the authors for adding new analysis and also for changing some of the discussion and conclusions of the results. Understandably, further experiments suggested in mouse ESCs are not feasible.

Data shown in this manuscript clearly shows inhibition of transcription leads to reduced acetylation and transcription contributes to histone acetylation patterns.

This version is suitable for publication in Nature communications.

Thank you for your comments and for your understanding.

Minor points: Authors should discuss recent publication in Genome research <http://www.genome.org/cgi/doi/10.1101/gr.257576.119>, which shows increased histone acetylation leads to increased binding of TFs and transcription.

This manuscript, entitled “Distinct contributions of DNA methylation and histone acetylation to the genomic occupancy of transcription factors”, by Cusack et al., 2020, shows that treatment of cells with a protein deacetylase inhibitor (TSA) increases DNA-binding of three of five transcription factors, suggesting that histone acetylation functions “upstream” of transcription. We respectfully disagree that discussion of this paper will add to our manuscript for the following reasons:

1. The results of Cusack et al., are not contrary to our work and we acknowledge the impact that acetylation might have on transcription in the manuscript. On lines 12-15 on page 14, we state “histone acetylation is proposed to facilitate transcription by directly modulating histone-DNA contacts or by targeting chromatin remodelers to disrupt nucleosomes. Thus, our research suggests that acetylation is a component of a feed-forward loop that maintains expression of active genes.”
2. Cusack et al. did not conclusively show that TSA promotes transcription factor binding through increased histone acetylation. As stated on lines 4-6 on page 3 in our manuscript, “in addition to histones, HATs have been shown to acetylate many non-histone proteins involved in transcription”. Cusack et al. did not rule out the possibility that TSA facilitates transcription factor binding through promoting acetylation of non-histone substrates. Indeed, acetylation of NRF1, one of the factors examined in the paper, promotes its interaction with DNA (PMID 12720548).
3. We are already over our recommended number of citations and thus cannot list all papers showing a positive impact of histone acetylation on transcription.

If, however, this reviewer is insistent that we include reference to Cusack et al., we will of course do so.

Please check grammar, e.g. Page 14 line 16

Our apologies and we have done a careful check of the manuscript for additional errors, which are highlighted by track changes.

Reviewer #3 (Remarks to the Author):

The authors have done an excellent job addressing the comments from all reviewers. I find this revised manuscript to be rigorous and the conclusion well supported by the data.

Thank you.

We would again like to express gratitude toward the reviewers for their assistance in greatly improving our manuscript, and for their understanding regarding the challenges faced by my lab.

Best wishes,

LeAnn Howe
Department of Biochemistry and Molecular Biology
University of British Columbia
ljhowe@mail.ubc.cas